# Peer review of "Remodeling the ECM: Implications for Metastasis and Tumor Dormancy"

_cancers, 2021, doi:10.3390/cancers13194916_

Round 1

Reviewer 1 Report

In the present article entitled " Remodeling the ECM: implications for metastasis and tumor dormancy”, the authors have summarized the current knowledge of ECM and its role in metastasis and dormancy in Cancer. Furthermore, the authors have highlighted the importance of studying ECM in the context of stem cell quiescence and dormancy. In addition, the authors have raised some burning questions which need to be addressed in this regard. The present article is well written and provided all the present literature in a concise and precise way. However, the present article lacks a clinical translational perspective. Authors have raised this question in the discussion, however, presently a lot of drugs that are targeted to ECM, are under investigation and showed great promise in preclinical studies. Authors need to discuss the potential of those drugs in the context of metastasis and dormancy. Furthermore, some part of the article has different font then rest of the manuscript. Authors need to make sure all the text has a similar font.   

Author Response

REVIEWER 1

In the present article entitled " Remodeling the ECM: implications for metastasis and tumor dormancy”, the authors have summarized the current knowledge of ECM and its role in metastasis and dormancy in Cancer. Furthermore, the authors have highlighted the importance of studying ECM in the context of stem cell quiescence and dormancy. In addition, the authors have raised some burning questions which need to be addressed in this regard. The present article is well written and provided all the present literature in a concise and precise way.

Response: We thank the reviewer for her/his constructive comments that have strengthened the conclusions of our manuscript.

However, the present article lacks a clinical translational perspective. Authors have raised this question in the discussion, however, presently a lot of drugs that are targeted to ECM, are under investigation and showed great promise in preclinical studies. Authors need to discuss the potential of those drugs in the context of metastasis and dormancy.

Response: We have included a new paragraph with references highlighting drugs that target the ECM (page 9).

Furthermore, some part of the article has different font then rest of the manuscript. Authors need to make sure all the text has a similar font.   

Response: Thanks for noticing it. It has been corrected.

Reviewer 2 Report

This manuscript by Di Martino et al. has implications of the ECM of the cancer cells disseminate (DTCs) for dormancy ECM of the niche, and for their escape DTCs of remodeling the ECM.

Major Comments:

The authors state that the: “This knowledge will pave the path for the development of new therapies to eliminate dormant cells by targeting their ECM microenvironment. A strategy that could ultimately lead to permanent control over cancer relapse and hope for a treatment to prevent metastasis.” You state that: “Enriching in ECM components that promote dormancy will favor an anti-proliferative microenvironment that would prevent tumor cell awakening.” What do authors think of this? The authors will and they play in anti-tumors roles. In the patients with cancer will the DTCs in the tissue and “ECM molecule may play multiple roles in the context of dormancy depending on the localization and the tumor type studied.”

The immune system has polarization (anti-tumor and pro-tumor, but they are a gradient of these to extremes) of macrophages (in the Conclusions), neutrophils, and NK cells and they produced MMPs, proteases, TIMPs and growth factors (TGFbeta for M2-like macrophages). The authors must have section on the immune systems. TGFbeta is anti-tumor roles in the beginning, but the end of the pro-tumors roles, and TGFbeta has suppressive the immune system.

Tissue inhibitors of metalloproteases (TIMP) exert dual roles in cancer (TIMP1 the have pro-tumor roles, and patients have TIMP1 has poorly survival capability). I looked TIMP http://matrisomedb.pepchem.org/ I saw that MMPs, TIMP1, TIMP2 and TIMP3, in is the matrisome of the tumors. The authors must have section MMPs, proteases and TIMPs.

Minor Comments:

Proteomics-based studies of the ECM matrisome, you should cite: http://matrisomedb.pepchem.org/ and also cite: MatrisomeDB: the ECM-protein knowledge database  Xinhao Shao, Isra N Taha, Karl R Clauser, Yu (Tom) Gao, Alexandra Naba. Nucleic Acids Research, Volume 48, Issue D1, 08 January 2020, Pages D1136–D1144, https://doi.org/10.1093/nar/gkz849.

Author Response

REVIEWER 2

Response: We thank the reviewer for her/his constructive comments that have strengthened our manuscript.

Major Comments:

The authors state that the: “This knowledge will pave the path for the development of new therapies to eliminate dormant cells by targeting their ECM microenvironment. A strategy that could ultimately lead to permanent control over cancer relapse and hope for a treatment to prevent metastasis.” You state that: “Enriching in ECM components that promote dormancy will favor an anti-proliferative microenvironment that would prevent tumor cell awakening.” What do authors think of this? The authors will and they play in anti-tumors roles. In the patients with cancer will the DTCs in the tissue and “ECM molecule may play multiple roles in the context of dormancy depending on the localization and the tumor type studied.”

Response: We have included new reference to highlight recent drugs that target the ECM to treat tumors (page 9). We hypothesize that the same rationale could be used to target dormant cells.

The immune system has polarization (anti-tumor and pro-tumor, but they are a gradient of these to extremes) of macrophages (in the Conclusions), neutrophils, and NK cells and they produced MMPs, proteases, TIMPs and growth factors (TGFbeta for M2-like macrophages). The authors must have section on the immune systems. TGFbeta is anti-tumor roles in the beginning, but the end of the pro-tumors roles, and TGFbeta has suppressive the immune system.

Response: We appreciate the point raised by the reviewer but to adhere to the length of the perspective article we have decided to leave out certain areas that, while very interesting, will require an extensive development, such as the immune system.

Tissue inhibitors of metalloproteases (TIMP) exert dual roles in cancer (TIMP1 the have pro-tumor roles, and patients have TIMP1 has poorly survival capability). I looked TIMP http://matrisomedb.pepchem.org/ I saw that MMPs, TIMP1, TIMP2 and TIMP3, in is the matrisome of the tumors. The authors must have section MMPs, proteases and TIMPs.

Response: We have included a new paragraph discussing MMPs in dormancy (page 6).

Minor Comments:

Proteomics-based studies of the ECM matrisome, you should cite: http://matrisomedb.pepchem.org/ and also cite: MatrisomeDB: the ECM-protein knowledge database  Xinhao Shao, Isra N Taha, Karl R Clauser, Yu (Tom) Gao, Alexandra Naba. Nucleic Acids Research, Volume 48, Issue D1, 08 January 2020, Pages D1136–D1144, https://doi.org/10.1093/nar/gkz849.

Response: We have included this reference.
